# Emergency Primary Ureteroscopy for Acute Ureteric Colic—From Guidelines to Practice

**DOI:** 10.3390/jpm12111866

**Published:** 2022-11-08

**Authors:** Yasmin Abu-Ghanem, Christina Fontaine, Radha Sehgal, Luke Forster, Neeta Verma, Gidon Ellis, Rajesh Kucheria, Darrell Allen, Paras Singh, Anuj Goyal, Leye Ajayi

**Affiliations:** Department of Urology, Royal Free London NHS Foundation Trust, London NW3 2QG, UK

**Keywords:** ureteric stones, temporizing measures, ureteric stent, definitive treatment, ureteroscopy, extracorporeal shock wave lithotripsy, guidelines

## Abstract

Objective: To review the factors that may influence the ability to achieve the present guidelines’ recommendations in a well-resourced tertiary centre. According to current National Institute for Health and Care Excellence (NICE) guidelines, definitive treatment (primary ureteroscopy (URS) or shock wave lithotripsy (ESWL)) should be offered to patients with symptomatic renal colic that are unlikely to pass the stone within 48 h of diagnosis. Methods: Retrospective review of all patients presenting to the emergency department between January and December 2019 with a ureteric or renal stone diagnosis. The rate of emergency intervention, risk factors for intervention and outcomes were compared between patients who were treated by primary definitive surgery vs. primary symptom relief by urethral stenting alone. Results: A total of 244 patients required surgical management for symptomatic ureteric colic without symptoms of urinary infection. Of those, 92 patients (37.7%) underwent definitive treatment by either primary URS (82 patients) or ESWL (9 patients). The mean time for the procedure was 25.5 h (range:1–118). Patients who underwent primary definitive treatment were likelier to have smaller and distally located stones than the primary stenting group. Primary ureteroscopy was more likely to be performed in a supervised setting than emergency stenting. Conclusions: Although definitive treatment carries high success rates, in a high-volume tertiary referral centre, it may not be feasible to offer it to all patients, with emergency stenting providing a safe and quick interim measure. Factors determining the ability to provide definitive treatment are stone location, stone size and resident supervision in theatre.

## 1. Introduction

Urolithiasis is one of the most common urological conditions. In the past several decades, the prevalence of urolithiasis has increased dramatically, reaching a lifetime frequency of 14% [1,2]. The rising prevalence is often associated with increased hospital visits, investigations and interventions, all of which pose a significant financial burden and make cost-efficient management of these patients imperative [3].

Traditionally, primary management of ureteric stones included temporizing measures such as analgesia, and expectant or late management. When an intervention is indicated, like severe kidney injury, an infected stone or ongoing pain, primary treatment is often given in the form of a ureteric stent or nephrostomy insertion to relieve the patient’s symptoms before definitive treatment [4,5,6]. However, these measures can result in frequent hospital visits with further symptoms, additional procedures, delays in treatment and possible complications, thus worsening the burden of urolithiasis on both the patient and the healthcare system.

Subsequently, in the last few years, accumulating clinical and financial evidence supports the definitive management of ureteric stones, with primary ureteroscopy (URS) or extracorporeal shock wave lithotripsy (ESWL) as reasonable first-line treatment options for ureteric stones to avoid long-term stenting, whenever appropriate [6,7,8,9,10].

This practice has been recently supported by the National Institute for Health and Care Excellence (NICE) guidance. According to the NICE guidelines, a definitive primary treatment (by URS or ESWL) should be offered within 48 h of diagnosis to patients with ureteric stones that are unlikely to pass and intractable pain [5].

However, despite promising results, the number of studies that have examined the fundamental role of primary URS in the UK or other public healthcare-dominated systems is limited.

The present study aims to assess whether the NICE strategy is implemented in daily clinical practice and to examine the factors that may influence achieving these guidelines in a well-resourced tertiary teaching hospital.

## 2. Materials and Methods

This is a retrospective analysis of all patients requiring emergency intervention for our institution’s computed tomography (CT)-confirmed ureteric calculus. All patients underwent either ureteric stenting, primary URS or ESWL between January and December 2019. Indications for intervention included: persistent pain despite adequate analgesic medication, persistent obstruction and renal insufficiency. Patient demographics and operative details were collected retrospectively by reviewing electronic records, including medical notes, operation notes and discharge summaries. The following demographic data were recorded: age, gender, stone size, location, time of admission, type of intervention, the time between diagnosis and intervention (time to intervention—TTI), length of hospital stay, presence of consultant during the procedure and number of re-admissions following the initial procedure. Time of admission was defined as either “in hours”—between 08:00 and 17:00, or “out of hours”—between 17:00 and 08:00.

Successful treatment was defined as a complete primary treatment (ureteroscopy or ESWL) without the need for additional intervention other than stent removal. All patients underwent CT scan 6 months following the definitive procedure to ensure their stone-free status. The number of re-visits was defined as reattendances to the accident and emergency department (A&E) during the following three months after the initial visit.

### 2.1. Stent Insertion

Retrograde ureteric double-J stent insertion was performed under general anaesthesia. All stents were inserted using a cystoscope with a 30° lens and guided by a fluoroscope. A cystoscope was introduced to the desired ureteral orifice, and a Termo glide guidewire (0.035 inches) was inserted into the ureteral orifice, behind the stone up to the kidney. The stent was placed next over the guidewire and pushed into the kidney using a pusher. Stent size was selected on an individual patient basis. Ureteric stents were either removed during definitive surgery or in clinic by flexible cystoscopy under local anaesthetic if the patient’s stone had passed.

### 2.2. Ureteroscopy

All emergency primary URS was undertaken with the intent to provide definitive treatment. URS was performed under general anaesthesia with an 8F semirigid ureteroscope (Karl Storz Endoskope, Tuttlingen, Germany) with the aid of fluoroscopy. Stones were either fragmented using a holmium:YAG or removed by an endoscopic basket. Any stone fragmentation was performed with a holmium:YAG laser (Lumenis Ltd., Elstree, UK). When required, stone fragments were removed using an endoscopic basket. The decision to place a ureteric stent following the procedure was left to the operating surgeon’s discretion.

### 2.3. ESWL

ESWL was performed by the same dedicated radiographer in all cases using an on-site lithotripter (Storz Medical Modulith SLX-F2). The number of delivered shocks varied depending on stone size and density, up to 3000 pulses. Maximum shockwave energy and speed delivered were 7J and 4 Hz, respectively.

### 2.4. Statistical Analysis

Data are presented as mean ± standard error and range, or number (percent) unless otherwise specified. Statistical analysis was performed using Statistical Package for Social Sciences (SPSS, Version 22.0, Chicago, IL, USA). The student’s *t*-test and the Mann–Whitney U test were used for the analysis of continuous variables and the Chi-square test was used for the analysis of categorical variables. A *p* value of <0.05 was considered statistically significant.

## 3. Results

A total of 287 consecutive patients underwent emergency intervention for ureteric calculi. Of those, 244 were included in this study and 13 were excluded. Primary definitive treatment was performed in 92 patients (37.7%), including URS in 83 (90.2%) and ESWL in 9 (Figure 1). Overall, 92 (37.7%) patients underwent primary treatment, whereas the remaining 152 had surgical stenting without final stone extraction.

The baseline parameters of the groups are seen in Table 1. Both groups were comparable in regards to age and gender. However, patients who underwent primary definitive treatment were more likely to have smaller and distally located stones than the primary stenting group (Figure 2). Moreover, although the time of admission (in or out of office hours) did not seem to affect the type of treatment chosen, a significant association was found between consultant presence in theatre and the type of procedure performed. A consultant urologist was present during the operation in 22.4% of ureteric stenting cases compared to 68.7% of definitive emergency procedures (68.7%). The rest of these procedures were performed by a urology registrar alone.

Nevertheless, no differences were observed in the TTI as in both groups, surgical intervention was delivered in up to 48 h (30.7 and 25.3 h for primary stenting and definitive treatment, respectively). Of note, the hour of procedure was not associated with the presence of a consultant. A consultant was present in most cases during the “out office hours” (62.6% of all procedures and 57.5% of URS).

Given the potential inherited bias between patients undergoing ESWL or URS, we excluded the patients who underwent primary ESWL and compared the primary stenting group to primary URS alone (Table 1). The second analysis revealed similar results, including smaller and more distally located calculi in the primary URS group compared to the primary stenting one. Notably, we found that all ESWL procedures were performed without the presence of a consultant urologist.

### Surgical Outcome

In terms of surgical outcomes, the overall success rate was 83.7% and 97.4% for primary stone treatment and primary stenting, respectively. Treatment failures management is specified in Table 1.

All patients who underwent primary ESWL achieved complete stone clearance. Successful primary URS was performed in 68 patients (82%). A ureteric stent was inserted at the end of all 68 primary URS. Reasons for treatment failure are shown in Table 1. Of the failed procedures, almost 50% were secondary to difficulty in access and a narrow ureteric lumen.

Further analysis did not reveal differences between the failed and successful procedures in regards to the age of the patients, size of the stone, time of admission or presence of a consultant during surgery. Nevertheless, patients who underwent successful URS were more likely to have distal stones than those who failed procedures (66.2% vs. 40%, *p* = 0.051) (Table 2).

Further outcome analysis was performed comparing the successful treatment groups (Table 1). The length of stay following surgery was comparable in all groups. However, patients who underwent stenting alone have a higher rate of A&E re-visits. All re-visits were due to stent-related symptoms (pain or urinary symptoms).

## 4. Discussion

Emergency URS or SWL provide feasible options for definitive treatment in symptomatic renal colic with comparable stone-free rates to elective treatment. However, emergency stenting also provides a safe and quick interim measure. The value of temporary procedures is mainly to shift urgent stone conditions to non-urgent pathways, allowing for prolonged drainage until the elective procedure. The advantage of this is seemingly straightforward. The patient arrives at a pre-set date and is potentially operated on by an experienced endourology team.

Nevertheless, pain relief by stent alone will also lead to a subsequent burden on elective and outpatient waiting lists, which will cause severe delays and potentially more associated complications like stent pains, forgotten stents and infections [11]. This has been compounded during the COVID-19 pandemic with a loss of elective operating, leading to more surgical delays [12,13]. Indeed, taking all of the above into consideration, the NICE guidelines have recently recommended applying primary definitive treatment whenever feasible. Several studies strongly supported a recommendation, including prospective comparative trials. However, in the current analysis, we tried to show whether this approach is adopted in the actual clinical setting of a tertiary teaching centre and not within the limits of a controlled study, where various everyday factors are coordinated and controlled.

Our analysis showed that the rate of primary definitive treatment is still 37.7%. The majority of potentially suitable patients were still stented first. However, further analysis revealed that the “choice to treat” was influenced by stone and setting-related criteria.

Regarding the stone variables, it has been previously described that the stone’s size and location both predict the stone-free rates following URS, as well as the complexity of the procedure [14,15,16]. Consequently, we have shown that patients with larger and more proximal stones were more likely to be stented rather than receive definitive treatment. Further analysis of the “failed URS” group revealed that the size of the stone was not associated with failure. Hence, size alone should not necessarily be a reason to avoid primary URS. Distally located calculi, on the other hand, were more likely to end up in success. Another interesting finding was the effect senior staff availability might have on practice. In the current analysis, primary URS was more likely to be performed in a supervised setting, yet the time of admission (in or out of office hours) was not predictive of any treatments.

Moreover, the presence of an experienced surgeon did not affect the success rate of URS. These findings could be explained simply by earlier reports, suggesting that less experienced surgeons have been suggested to have more complications and lower success rates [17,18,19]. Taking that into consideration, it seems that the presence of a fully trained, experienced surgeon has tilted the scale toward a complete procedure rather than a stent alone. Of note, the fact that the hour of procedure did not affect the results means that the choice is more driven by the reluctance of the trainee to complete the procedure alone.

We acknowledge the apparent bias in the current analysis, rising from the retrospective nature of the study and the relatively small number of failed procedures. We also realize that it is impossible to conclude, without all the relevant data, what may have led the surgeon to choose one approach over the other. However, the current study is a practical view of the implication of the guidelines and associated evidence. Despite strong support from previous trials, primary URS is still attempted in less than 40% of patients, even in a tertiary well-experienced centre.

## 5. Conclusions

Despite the potential benefits and the relatively high success rate reported, primary definitive treatment is yet to become everyday practice. According to the current study, factors determining the ability to provide definitive treatment are stone location, stone size and resident supervision in theatre. Primary treatment, and more specifically, primary URS, should be encouraged.

## Figures and Tables

**Figure 1 jpm-12-01866-f001:**
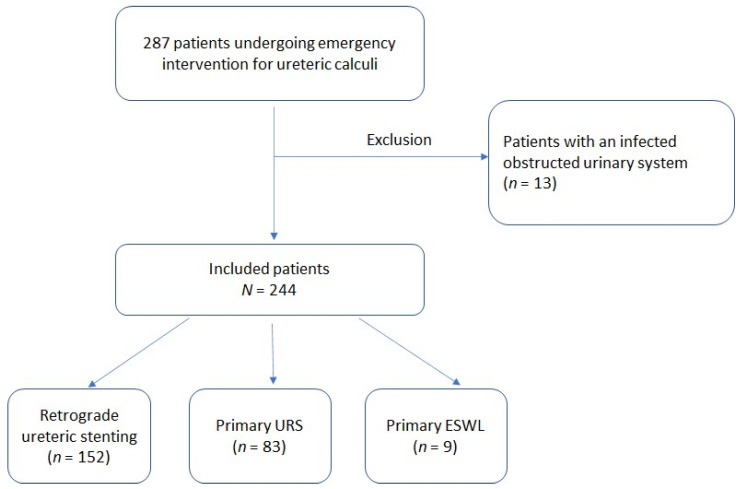
Flowchart demonstrating inclusion and exclusion criteria for the present study.

**Figure 2 jpm-12-01866-f002:**
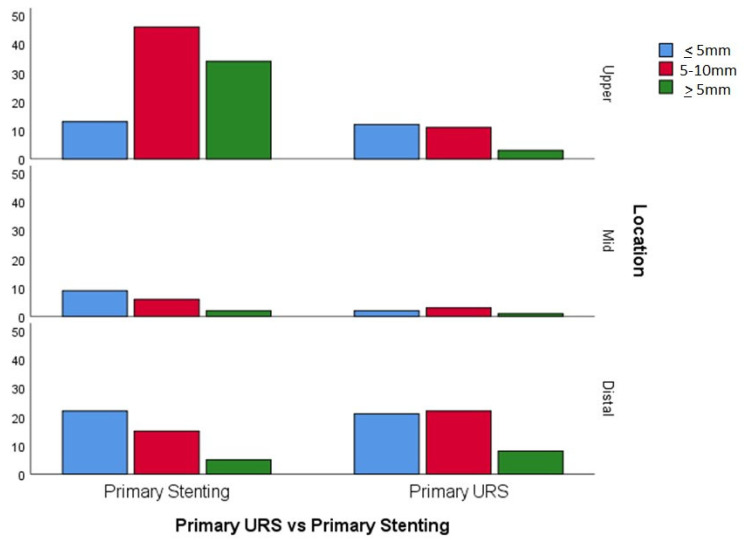
Type of treatment stratified by stone size and location.

**Table 1 jpm-12-01866-t001:** Univariate analysis of predictive factors for the different primary treatments types.

Variable	Primary Stenting (*n* = 152)	Primary Definitive Treatment (*n* = 92)		Primary URS (*n* = 83)	*p* Value
Age (mean ± SD)	52.1 ± 16.7	50.56 ± 17.1	0.49	51.5 ± 17.2	0.81
Gender					
Male	97 (63.8)	64 (69.6)	0.36	57 (68.7)	0.454
Female	55 (36.2)	28 (30.4)		26 (31.3)	
Size					
mean ± SD	7.76 ± 3.47	6.8 ± 3.12		6.92 ± 3.2	
			0.026		
≤5 mm	44 (28.9)	39 (42.2)		35 (42.2)	0.038
5–10 mm	67 (44.1)	40 (43.5)		36 (43.2)	
≥10 mm	41 (27)	13 (14.1)		12 (14.5)	
Location					
Proximal	93 (61.2)	30 (32.6)	<0.001	26 (31.3)	<0.001
Mid	17 (11.2)	7 (7.6)		6 (7.2)	
Distal/ VUJ	42 (27.6)	55 (59.8)		51 (61.4)	
Time of admission					
Office hours	57 (37.5)	39 (42.4)	0.448	34 (41)	0.602
Non office hours	53 (57.6)	95 (62.5)		39 (59)	
Consultant present in theater	34 (22.4)	57 (68.7)	<0.001	57 (68.7)	<0.001
Time from CT to Theater (hours)	30.7 ± 23.8	25.3 ± 22.1	0.078	25.9 ± 22.8	0.14
Treatment failure (and reason)	4 (2.6)			15 (18)	
	Difficulty advancing guidewire (*n* = 3)	PCN (*n* = 4)		Tight ureter (*n* = 7)	Stent (*n* = 6) PCN (*n* = 1)
	Inability to locate ureteric orifice (*n* = 1)			Proximal migration (*n* = 3)	Stent (*n* = 3)
				Residual fragments (*n* = 3)	Stent (*n* = 3)
				Other f* (*n* = 2)	Stent (*n* = 1) PCN (*n* = 1)
LOS	3 ± 4.48	2.5 ± 4.56	0.449	2.6 ± 4.5	0.15
Re-visits					
0	109 (73.6)	67 (87)		58 (85.3)	
≥1	39 (26.4)	10 (13)	0.021	10 (14.7)	0.05

LOS—length of stay; VUJ—Vesicoureteral junction; CT—Computed Tomography; PCN—percutaneous nephrostomy; * bleeding, poor vision, unable to access stone.

**Table 2 jpm-12-01866-t002:** Univariate analysis of comparing successful and failed URS.

Variable	Failed URS (*n* = 15)	Successful URS (*n* = 68)	*p* Value
Age (mean ± SD)	55.87 ± 17.6	50.6 ± 17.1	0.289
Gender			0.85
Male	10 (66.7)	47 (69.1)	
Female	5 (33.3)	21 (30.9)	
Size			0.783
mean + SD	6.87 ± 3.46	6.94 ± 3.2	
≤5 mm	7 (46.7)	28 (41.2)	
5–10 mm	6 (40)	30 (44.1)	
≥10 mm	2 (13.3)	10 (14.7)	
Location			0.051
Proximal	6 (40)	20 (29.4)	
Mid	3 (20)	3 (4.4)	
Distal/VUJ	6 (40)	45 (66.2)	
Time of admission			0.93
Office hours	6 (40)	28 (41.2)	
Non office hours	9 (60)	40 (58.8)	
Consultant present in theater	13 (86.7)	44 (64.7)	0.097

## Data Availability

Not applicable.

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
