# Peer review of "Emergency Primary Ureteroscopy for Acute Ureteric Colic—From Guidelines to Practice"

_jpm, 2022, doi:10.3390/jpm12111866_

Round 1

Reviewer 1 Report

Interesting idea but I have some remarks

1)The major concern about primary lithotripsy is complications. It is well documented that before URS urinary exams and cultures must be performed for urinary infection (that may not be apparent at the first diagnosis). If you go for lithotripsy at the primary setting then the infectious complications and also the ureteric trauma may be more increased than in the secondary setting. We do not see in this paper the complication rates and more specifically the infectious ones. Also we dont not see a more extended follow up in order to see if more strictures have been reported in the primary lithotripsy setting. 

2)The extended presence of a more experienced surgeon in the lithotripsy group may affect overall outcomes. 

3)What kind of pre operative planning did authors follow? The same with post operative follow up. How do authors define operative success in the lithotripsy group?

Author Response

We completely agree with all the important points raised by the reviewer. Nevertheless, the main objective of the current study is to review if current guidelines are being followed and of not why. What are the main factors that may influence the ability to achieve present guidelines recommendations in a well-resourced tertiary center. The objective was to review the decision making and the choices made based on clinical criteria. We acknolage the limitation of a retrospective observational study but we believe that the message is important in showing that not one size fits all and that guidelines may need to be tailored better. We also think that a retrospective study with a long term follow up is crucial and may help understand complications associated with any of the choices.

In regard to pre-operative planning, it is a retrospective study, hence there was no designed planning. Nonetheless, all cases were done in the same well-resourced tertiary center in which all surgeons followed the same standards as guided and required by BAUS and NICE. Successful treatment was defined as a complete primary treatment (ureteroscopy or ESWL) without the need for additional intervention other than stent removal. All patients underwent CT scan 6 months following the definitive procedure to ensure their stone-free status. A comment was added to the text. 

Reviewer 2 Report

Surgical outcomes section: could you specify how treatment failures were managed (i.e. nephrostomy tube or ureteral stent placement)?

Author Response

We thank the reviewer for this important comment. Treatment failure management was added to table 1.

Reviewer 3 Report

Dear Dr.Leye Ajayi

Your study needs ethical approval, including the registration number. Also, moderate English changes required

Author Response

Thank you for your comment. We conducted professional English editing and proofreading. This is an observational study that was conducted as an audit in accordance with the Clinical audits and service evaluations requirements.